# Attitudes towards Technology: Insights on Rarely Discussed Influences on Older Adults’ Willingness to Adopt Active Assisted Living (AAL)

**DOI:** 10.3390/ijerph21050628

**Published:** 2024-05-15

**Authors:** Ulrike Bechtold, Natalie Stauder, Martin Fieder

**Affiliations:** 1Institute of Technology Assessment, Austrian Academy of Sciences, 1010 Vienna, Austria; 2Department of Anthropology, University of Vienna, 1030 Vienna, Austria

**Keywords:** ageing in place, AAL adoption, major events in life, older adults, technology acceptance

## Abstract

Background: European research policy promotes active assisted living (AAL) to alleviate costs and reach new markets. The main argument for massive investments in AAL is its potential to raise older adults’ Quality of Life and enhance their freedom, autonomy, mobility, social integration, and communication. However, AAL is less widely spread in older adults’ households than expected. Research Aim: We investigate how the variable “technology acceptance” is connected to socio-economic-, social, health, “personal attitude towards ageing”, and “Quality of life” variables. Method: We conducted a study in Vienna between 2018 and 2020, questioning 245 older adults (M = 74, SD = 6.654) living in private homes. We calculated multivariate models regressing technology acceptance on the various exploratory and confounding variables. Results: Experiencing an event that made the person perceive their age differently changed the attitude towards using an assistive technological system. Participants perceived technology that is directly associated with another human being (e.g., the use of technology to communicate with a physician) more positively. Conclusion: Older adults’ attitudes towards technology may change throughout their lives. Using major events in life as potential entry points for technology requires awareness to avoid reducing the lives of older adults to these events. Secondly, a certain human preference for “human technology” may facilitate abuse if technology is given a white coat, two eyes, a nose, and a mouth that may falsely be associated with a natural person. This aspect raises the ethical issue of accurate information as a significant precondition for informed consent.

## 1. Introduction

### 1.1. Background: Active Assisted Living (AAL) Genesis and Research Policy Logics

AAL includes technology-based tools, devices, and services that facilitate ageing in place by helping older adults remain mobile and keep in touch with their social surroundings. Applications to monitor or improve older adults’ health, security, and safety are also at the heart of AAL.

Confronted with the rapidly changing demographic structure in Europe (and the World), European decision-makers strongly promoted technology in the last two decades as one way to alleviate the respective challenges [1,2,3,4]. So, in 2008, the Joint European Initiative AAL was launched to tackle the challenges of an ageing society and combine them with expanding market opportunities created by new technologies.

Four years after its implementation, the acronym AAL within the Joint European Initiative AAL was renamed from “ambient” to “active” assisted living. This indicates a shift from a relatively “passive connotation” of technology use to a more active one. Boudiny stated that this conceptualisation differs from healthy or productive ageing concepts [5]. By “fostering adaptability, supporting the maintenance of emotionally close relationships and removing structural barriers related to age or dependency”, it also embraces frail older adults. Van Dyk et al. (2013) criticise the politicisation of the “active ageing paradigm” and its likelihood of simplifying the target group into a homogenous group, similar to Bechtold et al., 2021 [6,7].

Therefore, the “active ageing” approach is likely to overlook its very target group’s diversity while ignoring that it assumes the responsibility of successful ageing to be in the hands of the individuals themselves [8].

Despite the long history in European research, these issues become visible as the dispersion of AAL applications in older adults’ households falls short of its potential [9,10,11,12,13].

### 1.2. Anthropological View on Ageing

Although there are different hegemonial role models in our society about age(s), it is evident that old age is as diverse as all other life stages [14,15]. Anthropologically speaking, physical and mental activities, individual biological imprints (e.g., genetic predispositions), and changes including historical, social, and societal influences and constructions are relevant here [16,17]. Sargent-Cox (2017) coins ageing as a fluid social construction that affects every individual: ageing in its natural form is a way of moving through different life spans [18]. Still, ageist attitudes lead to a divide between the “in-group” of youth and the “out-group” of old age. In addition to Sargent-Cox, Nägler and Wanka, 2022, propose that the different life stages and their transitions are not only a temporal phenomenon, but they are interwoven between each other by bodies, space, and things [19]. Moreover, actual self-perception is coined by negative stereotyping and physical performance [20,21,22]. A certain reluctance of older adults to deal with their age and ageing is often connected with negative images of old age and the tabooing of old age in our society [14,23]. The focus on ageing is often on the end of life; thus, the tendency towards repression is intensified [24]. However, a (collective) repression mechanism is also comprehensible here because it is biologically expected that the end of life may go hand in hand with frailty and a need for help [25].

This paper has a relatively broad research question precisely because of the multitude of relevant aspects of approaching older adulthood. Notwithstanding the seemingly universal effects of getting older, the persons involved are as diverse as other demographics [14,18]. We assume it is essential to consider the individual self-perception of ageing, which allows us to incorporate the diversity of ageing. These are prerequisites to tackle the role technology can play in older adults’ lives and what makes older adults more prone to adopt assistive technologies.

We investigate this by asking the following questions:What fosters their willingness to spend money and rearrange their everyday lives (change in habits and or surroundings) in favour of introducing an assistive technology system that they perceive to be useful?What is their attitude towards more invasive technologies (such as sensors or microchip implants)?

The overall research question is as follows: What are the relevant factors that influence older adults’ attitudes towards technology?

## 2. Method

### 2.1. The Questionnaire and Its Background

A comprehensive questionnaire was conducted between 2018 and 2020, covering a sample of 245 older adults (age between 61 and 93 years, M = 74, SD = 6.654). The target group consisted of self-reliant Viennese older persons living in their homes (receiving different amounts and forms of help). The questionnaire was divided into three parts, which cover the most critical influences on ageing and extend them to the context of older adults’ attitudes towards technology:Living conditions and socio-economic environment, social integration and state of health (incl. experiences of falling and hospitalisation), and perception of one’s own QoL [26,27,28].Willingness to undertake substantial changes in their everyday life or living environment and to invest in assistive technology in their homes, including the significance of AAL—active assisted living.Assessment of how their ageing is perceived (continuum/breaks) in itself and compared with others, as well as what their wishes and ideas are as they become older [29].

We used self-reported measurements on Quality of Life, health, and perception of own ageing, as these are reliable and valid tools [30]. We refer to Wang et al. (2018), who state that such measures capture older adults’ “(…) overall psychosocial situation” from their perspective and hence provide a “better methodological homogeneity and comparability of the condition of groups across studies and countries” [31]. A validation of the final questionnaire was drawn in 2016 (testing phase).

### 2.2. Framing of the Research Hypothesis

Potential technology acceptance can be described as a general inclination or aversion towards “the intention to use technology”, as Peek et al. (2014) put it [32]. We divided the sample into “technology optimists” and “technology pessimists” for further statistical analysis based on participants’ attitudes towards technology and how they influence or are influenced by other relevant factors (socio-economic status, social integration, Quality of Life, attitude towards more invasive technology, health of participants, and personal attitude towards ageing) as depicted in Figure 1.

*Technology optimists and technology pessimists* were defined by the questions “Imagine you are learning about an assistive technology system that seems appropriate for you and your household. (a) Would you be willing to spend money on it? (b) Would you be willing to change your daily habits (e.g., mealtimes) for it? (c) Would you be willing to change your accustomed environment (e.g., floor coverings) for it?” Possible answers to each question were “strongly agree”, “agree”, “disagree”, “strongly disagree”, and “I don’t know”. Those participants who showed a willingness to change (“strongly agree” or “agree”) on all three questions were labelled “technology optimists”. Those who indicated an unwillingness to change (“strongly disagree” or “disagree”) on at least one of the questions (to a maximum of all three questions) were labelled “technology pessimists”.

We assume a relation between technology optimists/pessimists and socio-economic status, social integration, Quality of Life and its potential benefits, attitude towards more invasive technology, health, and personal attitudes towards ageing. These factors are explained in more detail below.

Further, *social variables* were covered by questions about social integration (“Do you have regular contact—at least several times per month—with at least one person?”; “Do you have regular contact with at least one of your neighbours?”) and cohabitation.

*Quality of life (QoL)* is defined as the sum of an individual’s living conditions. It is measured along with the self-rated overall Quality of Life (oQoL).

The *impact that assistive technologies have on Quality of Life* was measured by asking participants to rate how technologies affect (a) their own QoL, (b) the QoL of their loved ones, (c) the QoL of (professional) caretakers, and (d) the Austrian health care—and social system.

*Participants’ health* was covered by self-rated health, the experience of physical limitations (e.g., walking problems, balance problems, vertigo, orientation problems, …), and experienced falls since age 55.

We also included their *attitude towards more invasive technology use*, specifically their interest in potentially communicating with their physician via television, computer, or tablet, their willingness to use sensors in their own homes (e.g., in the cutlery drawer, bedside rug, faucet, slippers, clock, and bathroom door) and their interest in using microchip implants in their arm that could measure medical parameters, such as blood pressure or blood sugar.

Finally, we were interested in participants’ *personal attitudes towards ageing.* This includes their self-perceptions compared to others in terms of their age regarding (a) health, (b) fitness, (c) activeness, and (d) contentedness. We asked participants if they felt that their calendrical age and their felt age matched; if there had been a specific event in time that made them feel older; if they sometimes felt older than they were and sometimes younger; if there was a particular birthday since which they felt older; if calendrical age was important to them; lastly, we also asked them what their felt age was (as opposed to their calendrical age).

### 2.3. Procedure and Sample

Older adults were approached by the authors predominantly in a personal manner, and of a total of 478 distributed questionnaires, only 135 were distributed without a personal face-to-face explanation beforehand. These received a group introductory or a written guide. Twenty-one organisations were approached as multipliers as well as natural places where the questionnaire was distributed (e.g., Vienna Pensioners Clubs, Viennese social services, an acute gerontology department of a Viennese Hospital group or Associations like “alters. kulturen”, and Vienna).

The questionnaire is anonymous, and we adapted the categories to foster privacy (e.g., we collected no postal codes but asked for the size of the place of residence). We kept no lists of persons or locations for gathering the questionnaires. Two hundred forty-five correctly filled questionnaires were given/sent back, making up the data basis. Participants were counted as missing variables if they did not answer certain questions. Therefore, the absolute number of answers for many questions in the “Results” section is variable.

We used SPSS IBM, version 27, R 4.0.5 (R Core Team 2020), libraries ordinal and psych for all analyses.

### 2.4. Multivariate Regression

Using a logistic regression (general linear model, R basis function) on the basis of the whole data set, we regressed the attitude of the use of a technological system surveyed by the question “I would be prepared to invest in a technological system and would change my habits and environment for it in order to support myself”, encoded as 0 = No and Yes = 1 (N = 135/84) separately on the following categorical social variables (cohabitation and social integration), QoL variables (Quality of Life and technology impact on QoL), health variables (self-rated health, physical limitations, and falls), attitude towards invasive technology, attitude towards ageing, controlling for age and sex (men = 1; female = 2), and education as a continuous variable (encoded as follows: 1: compulsory school; 2: vocational or commercial school; 3: high school; 4: university-related education; 5: university degree) on basis of a binomial error structure. The interpretation of “*technology optimists*” vs. “*technology pessimists*” is a posteriori interpretation after the data inspection.

## 3. Results

### 3.1. Descriptive Statistics

#### 3.1.1. Socio-Economic Factors and Social Integration

The sample consists of 151 women (61.6%) and 91 (37.1%) men (N = 242). The average age of male participants was 74 years (M = 73.6 SD = 6.601 N = 73), while the average age of female participants was 75 years (M = 74.6 SD = 6.682 N = 126).

A total of 15.5% (38) of participants completed compulsory school, 37.1% (91) completed vocational or commercial school, 15.5% (38) attained a high school diploma, 11.4% (28) completed some form of university-related education, and 20% (49) of participants attained a college degree (N = 244).

In total, 38.9% (93) of the participants had a net monthly income of EUR 1000–2000 at their disposal, 24.3% (58) had a net income of EUR 2000–3000, 10% (24) had EUR 500–1000, 8.4% (20) had EUR 3000–4000, 3.3% (8) had EUR 4000–5000, another 3.3% (8) had more than EUR 5000, and 2.9% (7) had less than EUR 500 (N = 239).

Approximately half of the respondents (51.6% or 126 persons) reported living with at least one person (N = 244). Moreover, 10.3% (25) of participants were single, 40.7% (99) were married and living with their spouse, 2.9% (7) were married and living separately from their spouse, 27.6% (67) were widowed, and 18.5% (45) of participants were divorced (N = 243).

A total of 43.1% of participants met their most important contact persons daily, 29.3% several times per week, and 31% several times per month (N = 239). In addition, 74.7% (180) maintained an active relationship with one or more of their neighbours (N = 241).

#### 3.1.2. Overall Quality of Life and Quality of Life Technology Benefits

In total, 82.9% (203) reported a good QoL, while 17.1% (42) reported an intermediate (or bad) QoL (N = 245). How technology to support ageing is perceived to affect their own and others QoL is summarised in Table 1.

#### 3.1.3. Use of Technology

Combining the answers for willingness to spend money, change habits, and change the environment for a new technological system, 38.4% (84) of the participants answered all three questions with “agree” and 61.6% (135) disagreed on at least one of the three questions (N = 219). Table 2 depicts the way sensors are perceived in the home. In Table 3 the concrete behavior in the course of acquiring technology at home is described and Table 4 summarises how respondents rated using technology (TV, computer, and tablet) to communicate with their physician and the use of a chip to measure medical parameters and regularly transmit these to their doctor.

#### 3.1.4. Health

More than half (51.4% or 126 persons) of the participants reported being satisfied with their health, and 18.8% (46) were very satisfied; 15.1% (37) were neutral towards their health, 10.6% (26) were unsatisfied, and 4.1% (10) were very unsatisfied with their health (N = 245).

The most commonly reported physical limitations were walking difficulties (44.5%, 109 persons, N = 222), followed by vision impairments (36.7%, 90 persons, N = 214), hearing impairments (36.7%, 90 persons, N = 215), and memory difficulties (32.7%, 80 persons, N = 213). Less frequently experienced limitations were vertigo (23.7%, 58 persons, N = 207), balance problems (19.2%, 47 persons, N = 200), and orientation problems (9%, 22 persons, N = 202).

In total, 71.4% of participants reported one or more falls since age 55, while 28.6% never experienced this event (N = 234).

#### 3.1.5. Perspective on Getting Older

The perceived age was, on average, 65.62 years (SD = 9.912, N = 198). This seems about ten years lower than the actual age (M = 74.27 years, SD = 6.654, N = 199). Differences between perceived and chronological age show that 130 respondents felt younger, whereas only nine felt older than their actual age. Twenty-five people reported a match of perceived and chronological age.

When compared to others, respondents rate themselves higher than their peers. Specifically, 51% feel healthier (N = 224), 53.9% fitter (N = 230), 62.4% more active (N = 232), and 62.9% more content (N = 226). Table 5 summarises how major life events connected to their (perceived) age.

### 3.2. Multivariate Regression

We found that a specific event that made a person feel older also positively altered their attitude towards using a technological system (see Table 6). The ODD ratio suggests a stronger influence on attitude change than the other explanatory factors. Thus, age, gender, education, and income have no significant effect on a person’s attitude towards using a technological system, and the odds ratios of these variables are fairly close to 1. The significant negative association between “feeling older since a certain event” and the willingness to actively use assistive technology is due to the fact that feeling older after a life event is coded as 1 (not feeling older as 2) and “being open to using assistive technology” is coded as 1 and 0 as no, so an event that makes someone feel older leads to greater acceptance of technology. Due to the same coding of “sensors” and “microchip”, a negative association appeared in the regression model, indicating that people who accept sensors and microchips are also more likely to accept technology in general.

In addition, we found that someone with a more negative attitude towards using technology (“technology pessimist”) also has a more limited attitude towards using a sensor or chip implant that records medical data. Again, all control variables remain non-significant (see Table 7 and Table 8).

Moreover, we found no significant effect for any other exploratory variables in the separate models (see Table 9). In Figure 2 the result is shown when the attitudes towards the use of a technological system (technology optimists and technology pessimists) are regressed on socio-economic variables (as captured by the covariates: sex, age, education, and income), social variables (captured by the variables: cohabitation and social integration), QoL variables (captured by the variables: Quality of Life and technology impact on QoL), health variables (captured by the variables: self-rated health, physical limitations, and falls), the attitude towards invasive technology, and attitude towards ageing.The latter two are significant.

## 4. Discussion

Peek et al. (2014) state that current technology acceptance models lack integration of specific factors concerning individuals, such as social embeddedness or actual physical conditions [32]. We integrated this by looking at older adults’ individual (social and physical) environments and incorporating “major events in life”. However, these commonly discussed factors, which are assumed to be relevant for technology adoption, such as social integration, health issues, subjective age, self-rated Quality of Life, or technology affordability, proved insignificant in our sample.

### 4.1. Major Events in Life and Technology and AAL Attitude

However, our findings show a significant positive association between the fact that they “feel older since a specific event” and the willingness to adopt assistive technology actively. In literature, this is known as *major events in life* [33]. Lindquist et al. (2016) describe these major events as hospitalisations, major falls, a diagnosis of dementia, the loss of a life partner, and an inability to keep up with activities of daily living [34]. Similarly, the SRRS (Social Readjustment Rating Scale) measures the impact of life events and includes items like death, separation, or increased number of marital arguments [35,36]. Ferguson et al. (2021) state that a health deficit can also be regarded as a major event in life, facilitating successful technology adoption, e.g., wearing a wearable that helps people with a particular disease [33]. Lindquist et al. (2016) argue that major events in life can cause different individual reactions, such as a more or less conscious denial of possible future changes or the delegation of their needs to someone else (mostly offspring) [34]. In addition to these findings, Banks et al. (2020) state that major events in life can lead to a systematic re-evaluation of older adults’ lives and their approach to tackling problems in their day-to-day lives [37].

In our case, major events in life and, consequently, the risk avoidance of older adults led to an increased willingness to potentially purchase or install an assistive technological home system that seemed helpful to them. At the same time, a certain reluctance to act in the face of major events in life by the affected persons may foster family members’ ability to introduce a new assistive technological system in times of change [34]. The importance of social surroundings on technology adoption has a long history in several models, including the Technology Acceptance Model (TAM) as well as its derivatives, e.g., UTAUT—Unified Theory of Acceptance and Use of Technology [38,39].

These results show that an individual’s perspective on the process of becoming older is essential and that this highly subjective perspective is dynamic. However, considering one’s own ageing is frequently overlooked when thinking about technology [40]. The phenomenon that as we grow older, our reasoning and motivation change has also been researched on a psychological basis: Carstensen (2021) argues using the framework of the Socio-emotional Subjectivity Theory that the subjective experience of how much time we have left is a significant factor for our motivations [41]. As we progress through life and become older, we move from an open-ended life experience early on to a more finite experience in older age. This framework also makes sense in our case: The very point in time when older adults see themselves as “old” and their life as increasingly finite is highly subjective, and major events in life play an essential role here. Also, how older adults perceive themselves plays an essential role, as does the positive framing of their ageing.

### 4.2. Technology Optimists and Pessimists Are Both Positively Triggered by Technology That Connects the User with a Real Person

Moreover, our findings show that about one-third of the participants were interested in investing money, changing their habits, and changing their environment for a new, helpful technological home system. The participants who were generally open towards adopting technology (technology optimists) were also more likely to accept more invasive technology like sensors in their homes and microchip implants to measure health status (e.g., blood sugar). The idea of observing technology adoption (patterns) has a long history, especially in the theory of the five stages of technology adoption, ranging from interested early adopters to a sceptical late majority [42].

Surprisingly, in our sample, both the so-called technology optimists and the technology pessimists showed high acceptance of communicating with their physicians via technology (in our case, a TV, computer, or tablet).

This is probably the case because of a deeply rooted (often unconscious) trust in human relations over trust in mere technology. A body of literature states that older adults consistently prefer robots to assist them with instrumental tasks (e.g., keeping the house clean) over robots that help with more intimate tasks (e.g., personal care). Gilleard and Higgs assume that this could be due to the greater vulnerability experienced in intimate care that wants to be experienced with an actual human [43,44,45]. Hence, technology that facilitates (real) human interaction could be seen by users as the best of both worlds. It facilitates social contact and reassurance from a real human being while it is easily accessible and a time-saving endeavour. However, users likely do not seem to find technology as such enticing, but they like the real human interaction it facilitates.

The fact that technology associated with human interaction is perceived more positively [46,47] may suggest that technology related to human interaction, even at the virtual level (e.g., humanoid appearing robots, humanoid user interfaces, and AI with a human face), leads to higher acceptance. In the field of intelligent personal assistants (IPAs, e.g., Alexa and Google Assistant), technology acceptance has been associated with the parasocial relationship theory, which describes the emergence of a one-sided bond between humans and human-like technology [48]. Another critical factor is interpersonal attraction, mainly task attraction, which is the emergent desire to work with someone or, in this case, some piece of technology [49]. These insights underline that the more human-like IPAs appear, the stronger the attachment to and the acceptance of IPAs. Regarding assistive technology for older adults, this kind of acceptance could occur more frequently among users who do not fully understand the mechanisms by which these robots operate.

Suppose the technology itself does not necessarily provide actual human support but merely imitates the humanoid appearance and/or interactions (e.g., an interface with avatars of medical staff or companionship robots of humanoid appearance). In that case, this could lead to a disguise of what assistive technology actually is and does. In a systematic review of wearable cardiac monitoring systems by Ferguson et al. (2021), one of the users liked the idea of “someone looking continuously “at their data and stated that this made him/her comfortable and fostered trust [33]. Therefore, the technology was attractive to them as it was associated with another human being. However, this is clearly different from the way such technologies typically work. Ursin, Timmermann, and Steger (2021) emphasise that the absence of an actual understanding of technology is a true ethical challenge for informed consent in the light of autonomy, beneficence, non-maleficence, and justice [50]. Informed consent requires transparency by linking understanding, autonomy (of decision,) and trust.

## 5. Conclusions

The fact that AAL is less widely spread in older adults’ households than one would think leads to the crucial question of what influences a person’s willingness to adopt a new assistive technological system (AAL). We find that the perception of AAL may differ depending on what assumptions we start with. Firstly, we find that the target group of older adults is dynamic, and attitudes towards technology are most likely not set in stone during a whole lifetime. Major events in life are inclined to raise older adults’ technology acceptance. Secondly, for both technology optimists and pessimists, it is relevant whether technological devices are associated with providing contact and/or support from another human being for its acceptance.

These results reveal two major challenges and require different ways to reduce potential negative impacts.

Our finding that major events in life are potentially raising older adults’ acceptance of changing their homes and introducing AAL aligns with the *challenge of simplifying or even paternalising a complex user group.*

Our results indicate that it seems advisable to look at possible major events in life, as technology acceptance seems lower if life proceeds in a perceived “smooth continuum”. However, we clearly emphasise the danger of being paternalistic here: technology producers should not use this information in a simple unidirectional deduction, reducing people’s lives to major events in life. Using fragile emotional set-ups would give rise to severe ethical issues and simplify the psycho-emotional state of individuals at specific points in time. Even though major life events seem to be a significant driver for technology adoption, it remains open to how this could be ethically implemented in marketing strategies and how it would reach its target audience. Moreover, apart from ethical considerations, people who have recently experienced blows of fate might not be open to advertising or even pay attention to it. Time matters in terms of timeliness: it may not be the catching of the event but the awareness of people being open to technology and change who previously acted rather reserved at some point in time.

2.Our finding that technology conveying real-life contacts within support structures is more likely to be accepted raises the challenge of differentiating a technology that merely simulates human support (by avatar, etc.).

The fact that technology pessimists also highly accepted a home-based e-health application in our sample could be interpreted as an indication that as soon as technology displays, involves, or uses a human face or touch (communication with an actual physician in our case), acceptance tends to be much higher than it would be without such a human interface.

However, this result poses a major ethical challenge if this knowledge is used in products to sell technology that only appears human but, in fact, is not. Here, the acceptance of “human” technology (technology that connects two or more real individuals) could be confused with a “human appearing” technology (a technology that disguises algorithms or AI and displays a human entity as the opposite of the user). In the latter case, users would need to be thoroughly informed and educated to avoid being misled by them.

This is in itself a severe ethical challenge, and it forms an ethical challenge for informed consent, which requires transparency by linking understanding, autonomy, and trust.

Hence the risk of misinterpreting acceptance of technology with a “human face” and thereby creating the pitfall of “false acceptance” needs to be taken into account consciously during the technology development process and in the marketing strategies. Regulations should ensure transparency about how the technology works and whether or not actual human assistance is provided.

## Figures and Tables

**Figure 1 ijerph-21-00628-f001:**
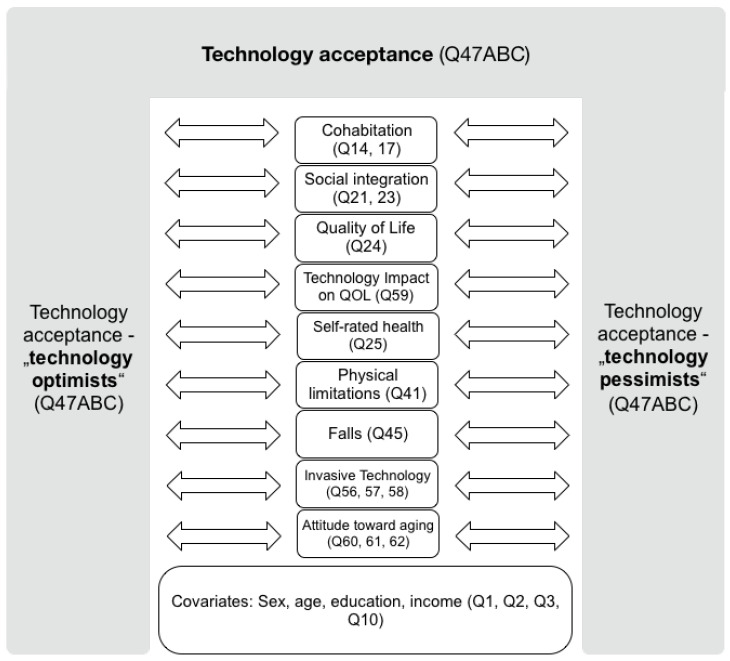
Methodological conceptualisation of a potential interrelation between technology acceptance of the questioned persons (they are split into “technology optimists” and “technology pessimists”, as indicated by the grey columns) and socio-economic variables (as captured by the covariates: sex, age, education, and income), social variables (captured by the variables: cohabitation and social integration), QoL variables (captured by the variables: Quality of Life and technology impact on QoL), health variables (captured by the variables: self-rated health, physical limitations, and falls), attitude towards invasive technology, and attitude towards ageing (created in “keynote” V8.1).

**Figure 2 ijerph-21-00628-f002:**
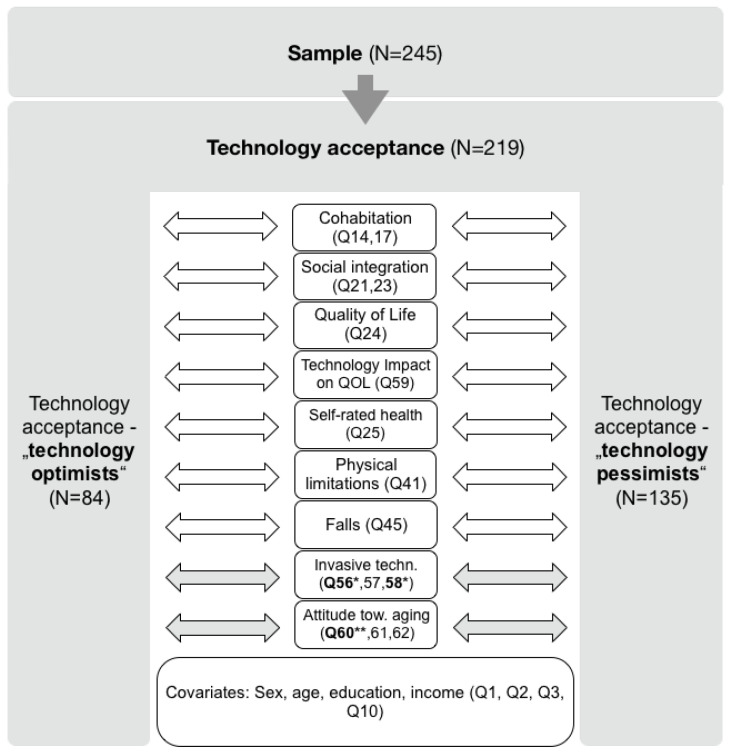
Attitudes towards the use of a technological system (technology optimists and technology pessimists) regressed on socio-economic variables (as captured by the covariates: sex, age, education, and income), social variables (captured by the variables: cohabitation and social integration), QoL variables (captured by the variables: Quality of Life and technology impact on QoL), health variables (captured by the variables: self-rated health, physical limitations, and falls), attitude towards invasive technology, and attitude towards ageing; significant results highlighted in bold with grey arrows (* < 0.05; ** < 0.01; *** < 0.001) (created in “keynote”, V8.1).

**Table 1 ijerph-21-00628-t001:** Respondents’ answers (in percentage and frequency) regarding the perceived impacts of how technologies affect them and their surroundings.

	Strongly Agree	Agree	Disagree	Strongly Disagree	Unsure
If older adults used technology to stay at home longer, it would increase their own Quality of Life (N = 227).	63.4% (144)	27.3% (62)	1.8% (4)	1.8% (4)	5.7% (13)
If older adults used technology to stay at home longer, it would increase the Quality of Life of their loved ones (N = 222).	49.1% (109)	32% (71)	5.9% (13)	3.2% (7)	9.9% (22)
If older adults used technology to stay at home longer, it would increase the Quality of Life of potential caretakers (N = 212).	41% (87)	29.2% (62)	6.6% (14)	4.7% (10)	18.4% (39)
If older adults used technology to stay at home longer, it would relieve the Austrian health care and social system (N = 226).	54% (122)	23.5% (53)	4.9% (11)	2.2% (5)	15.5% (35)

**Table 2 ijerph-21-00628-t002:** Distribution (in percentage and frequency) of the acceptance of sensors in various home objects.

		Yes	No	Not Specified
Could you imagine having a sensor installed in the following objects?	Cutlery Drawer (N = 208)	11.5% (24)	74% (154)	14.4% (30)
	Bedside Rug (N = 206)	7.3% (15)	79.6% (164)	13.1% (27)
	Faucet (N = 214)	29.9% (64)	57.5% (123)	12.6% (27)
	Shoes/Slippers (N = 205)	4.4% (9)	82.4% (169)	13.2% (27)
	Watch/Clock (N = 209)	19.6% (41)	65.6% (137)	14.8% (31)
	Bathroom Door (N = 212)	23.6% (50)	63.7% (135)	12.7% (27)

**Table 3 ijerph-21-00628-t003:** Distribution (in percentage and frequency) of openness and interest in using technology (TV, computer, and tablet) to communicate with their physician and to use a chip to measure medical parameters that would regularly be transmitted to their doctor.

	Yes	No	Not Specified
Could you imagine communicating with your doctor through your TV, computer or tablet? (N = 231)	54.1% (125)	39% (90)	6.9% (16)
If you said yes to the previous question, would you want this? (N = 134)	64.9% (87)	18.7% (25)	16.4% (22)
Could you imagine that your medical parameters (such as blood pressure or blood sugar) were measured by a chip in your arm and would be regularly transmitted to your doctor? (N = 230)	37% (85)	54.3% (125)	8.7% (20)

**Table 4 ijerph-21-00628-t004:** Respondents’ answers to their willingness to spend money, change their daily habits, and change their surroundings for a useful technical system in their household.

		Strongly Agree	Agree	Disagree	Strongly Disagree	Unsure
Imagine you find out about a technical system that you think would be useful for you and your household. How would you personally rate the following statements?	I would spend money on it (N = 227).	25.6% (58)	34.8% (79)	14.1% (32)	14.5% (33)	11% (25)
	I would be prepared to change my daily habits (e.g., mealtimes) for it (N = 215).	13.5% (29)	31.2% (67)	20% (43)	26% (56)	9.3% (20)
	I would be prepared to change my familiar surroundings (e.g., flooring) for it (N = 218).	13.3% (29)	28% (61)	23.9% (52)	27.5% (60)	7.3% (16)

**Table 5 ijerph-21-00628-t005:** Respondents’ answers to questions about their (perceived) age and life events connected to their (perceived) age.

	Strongly Agree	Agree	Disagree	Strongly Disagree	Unsure
My calendrical age and perceived age are the same (N = 229).	15.7% (36)	28.8% (66)	34.9% (80)	17.9% (41)	2.6% (6)
There was one specific event that made me feel much older than before (N = 224).	14.7% (33)	21.4% (48)	17% (38)	37.1% (83)	9.8% (22)
Sometimes I feel much younger and sometimes much older (N = 229).	28.4% (65)	28.4% (65)	22.7% (52)	15.3% (35)	5.2% (12)
Since a certain birthday, I have felt increasingly old (N = 228).	3.9% (9)	16.2% (37)	28.1% (64)	46.1% (105)	5.7% (13)
Age is not important to me (N = 227).	28.2% (64)	30% (68)	21.1% (48)	14.1% (32)	6.6% (15)

**Table 6 ijerph-21-00628-t006:** Attitude towards using a technological system in relation to “feeling older since a certain event”, controlled for age, sex, education, and income.

	Estimate	Std. Error	z Value	OR	*p*
(Intercept)	4.45572	2.494	1.787		0.074
**Feeling older since a certain event**	**−1.094**	**0.370**	**−2.953**	0.335	**0.003**
Sex	−0.18789	0.423	−0.444	0.829	0.657
Age	−0.05634	0.029	−1.927	0.945	0.054
Highest education	−0.04804	0.156	−0.308	0.953	0.758
Income	0.02216	0.198	0.112	1.022	0.911

**Table 7 ijerph-21-00628-t007:** Attitude towards the use of a technological system regressed on the potential use of a sensor, controlled for age, sex, education, and income.

	Estimate	Std. Error	z Value	OR	*p*
(Intercept)	2.207	2.228	0.991		0.322
**Sensor in home**	**−0.912**	**0.371**	**−2.461**	0.402	**0.014**
Sensor in home (no answer)	−0.244	0.641	−0.381	0.783	0.703
Sex	−0.110	0.406	−0.270	0.896	0.787
Age	−0.028	0.027	−1.051	0.972	0.294
Highest Education	−0.104	0.153	−0.678	0.901	0.498
Income	0.035	0.186	0.191	1.036	0.849

**Table 8 ijerph-21-00628-t008:** Attitude towards using a technological system regressed on the potential use of a chip implant, controlled for age, sex, education, and income.

	Estimate	Std. Error	z Value	OR	*p*
(Intercept)	3.8604923	2.384	1.619		0.105
**Microchip implant that takes medical parameters**	**−0.9019444**	**0.354**	**−2.550**	0.406	**0.011**
Sex	−0.0473046	0.418	−0.113	0.954	0.910
Age	−0.0529906	0.029	−1.837	0.948	0.066
Highest Education	0.0008788	0.156	0.006	1.001	0.996
Income	0.0082425	0.192	0.043	1.008	0.966

**Table 9 ijerph-21-00628-t009:** Supplementary data on other exploratory variables, such as parameter estimates and *p*-values.

	Estimate	*p*
Q14 Living alone	−0.227	0.52
Q17 Cohabitation	−0.046	0.91
Q21 Regular social contact	−1.125	0.17
Q23 Regular contact with at least one neighbour	0.407	0.26
Q24 Self-rate overall QoL	0.034	0.83
Q59A QoL benefit for self	0.542	0.53
Q59B QoL benefit for relatives	−0.159	0.79
Q59C QoL benefit for caretakers	0.106	0.85
Q59D QoL benefit for health and social system	−0.784	0.26
Q57 Talk to a physician with a TV, tablet, or computer	−0.188	0.63
Q41A Memory problems	−0.488	0.18
Q41B Walking problems	−0.293	0.43
Q41C Vision problems	−0.383	0.30
Q41D Hearing problems	0.108	0.77
Q41E Balance problems	−0.125	0.78
Q41F Vertigo	−0.623	0.14
Q41G Orientation problems	−0.315	0.54
Q25 Health satisfaction	−0.625	0.32
Q60A Perceived age and calendrical age coincide	−0.298	0.38
Q60C Sometimes I feel older, sometimes younger	−0.353	0.32
Q60D There is a specific birthday since which I feel older	0.153	0.72
Q60E Age is not important for me	−0.287	0.41
Q61A I feel healthier/less healthy than my peers	0.332	0.56
Q61B I feel fitter/less fit than my peers	−0.143	0.75
Q61C I feel more/less active than my peers	−0.385	0.40
Q61D I feel more/less content than my peers	0.046	0.95

## Data Availability

The datasets generated and analyzed during the current study are in the hands of the authors. Insights are available from the corresponding author on reasonable request.

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
