# Peer review of "Attitudes towards Technology: Insights on Rarely Discussed Influences on Older Adults’ Willingness to Adopt Active Assisted Living (AAL)"

_ijerph, 2024, doi:10.3390/ijerph21050628_

Round 1

Reviewer 1 Report

Comments and Suggestions for Authors

The paper addresses a significant problem – what influences the motivation of older adults to adopt active and assisted living technology.

The paper's first section starts with a historical overview of the field of AAL, followed by an anthropological view on aging and a resulting specification of research questions.    

The second section (method) describes the structure and application of a questionnaire, which constitutes the central tool of the presented research.    Subsection 2.2. explains the “framing” of the research and research hypotheses to be evaluated. The main aspects considered are technology acceptance, social variables, health-related aspects and personal attitudes toward aging. The section ends with information on the procedure, sample characteristics and statistical methods applied.

Section 3 – Results – starts with detailed information on the sample characteristics and the respondents' answers to the questions related to the different categories (technology, social, health, etc.). Specific questions / aspects are emphasized (e.g. significant life event, sensor in home, microchip implant), followed by an overview of results related to all aspects / questions.

The paper ends with a discussion (section 4) and conclusion (section 5) reflecting on the findings and connecting them to the related literature.

Review:

I agree with the general criticism expressed in the paper that the related research does not consider significant aspects (specifically social ones) appropriately. However, the approach presented in the paper also has some significant flaws on different levels. A related remark on the introduction – the provided definition of AAL is, in my opinion – too simplified. The respective technologies are designed to support independence and dignity in the home, not only/necessarily mobility and social connection in/to the outside world.    

Presentation Level:

The method section consists of a long and difficult-to-observe sequence of flowing text. I would prefer a tabular presentation of the results, where it is, for example, quickly observable how many (percentage) of the respondents agree with certain statements and to what extent (strongly agree – strongly disagree) and which are the percentual differences between specific questions/categories.

I missed a complete overview of the questions/statements in the paper. The related figures ( 1, 2) and Table 4 only contain a selection (in the “Q”-enumerations, some numbers are missing).

It would be of specific importance for me to see an overview of the technologies the participants were asked for. The range is big, from sensors in the environment (doors, drawers), body sensors (chip-implants, blood parameters) and communication technology (e.g. video conferencing with MD) and the respective devices have a different significance in terms of potential impact on daily life.  

Related literature level:

The authors implicitly/indirectly address related literature of different fields, but I believe an explicit reference/discussion of frequently referenced concepts would be necessary. One example is the TAM Model (and its derivates, e.g. UTAUT – indirectly covered by Peek et al.), which is extensively researched and also addresses aspects this paper focuses on. The same applies to technology adoption, which the authors label as “technology optimists” and “technology pessimists”. The related (and thoroughly researched) 5 stages of technology adoption model (innovators, early adopters, etc.) or the concept of digital natives/digital immigrants are not referenced (at least not directly). Finally, models addressing major life events (referenced by the Ferguson Model) have a long history, e.g. the Holmes and Rahe stress scale and several connections to AAL / technology acceptance. What is additionally missing is the impact/relevance of “daily hassles” in the context of technology, because it is not always the “big problems” that influence the adaption of technology, e.g. thematized in the Usability/User Experience (UX) related literature.   

So, my main criticism is not that the paper does not address a relevant topic but that comparing results with others in the field (and based on different models/concepts) is not appropriately reflected/has to be enhanced.        

Comments on the Quality of English Language

some typos, e.g. in the abstract (Vienna 2018-202), page 3 second bullet: Willingness TO undertake,

Author Response

Thank you very much for the review - we diligently considered all your suggestions and comments. 

Reviewer 2 Report

Comments and Suggestions for Authors

There are extensive issues with grammar/writing the makes the manuscript difficult to read. This is evident in the abstract, where there are many spelling/grammatical errors - " We conducted a study in Vienna (2018-202) to investigate how the variable “technology 11 acceptance” is connected to socioeconomic-, social-, health-, “personal attitude towards ageing”- 12 and “Quality of life”-variables) in older adults living at their homes (N=245, M=74, SD=6,654). " and "Mul- 13 tivariate models regressing technology acceptance on various explaining and confounding variables 14 showed that 1) major events in life 2) and a human in charge of the users' (health) data significantly 15 altered the attitude toward using an assistive technological system."

 It is also seems the analysis was highly selective with regards to the variables chosen, with some cherry picking of questions. There was no multiple comparison correction (such as a bonferroni's). The outcome variable is also a subjective one, instead of objective evidence of increased technological use. 

• Do you consider the topic original or relevant in the field? Does it 
address a specific gap in the field?

  No, as there are similar papers out there with better methodology
• What does it add to the subject area compared with other published 
material?   Very little, as the outcome variable was subjective.
• What specific improvements should the authors consider regarding the 
methodology?   The outcome measure should be an objective measure of technology use.
• Are the conclusions consistent with the evidence and arguments presented and do they address the main question posed? Please also explain why this is/ is not the case.
The analysis was highly selective with regards to the variables chosen, with some cherry-picking of questions. There was no multiple comparison correction (such as a bonferroni's). The outcome variable is also a subjective one, instead of objective evidence of increased technological use.  Comments on the Quality of English Language

Needs improvement.

Author Response

(The authors gave the same response as above.)

Reviewer 3 Report

Comments and Suggestions for Authors

The paper undertakes an interesting topic concerning attitudes of older persons towards technology, particularly to Active Assisted Living (AAL) solutions. The strength of the study is that the authors analyze many factors which can affect the attitudes to assistive technological systems as living conditions, socio-economic environment, social integration, self-related health, physical limitations, quality of life, perception of ageing. The study is limited to Vienna inhabitants and a relatively big sample of data was collected (245 correctly filled questionnaires).

I find the study and research questions interesting but certain things in the analysis are unclear and must be improved:

Line 11: it should be 2020 not 202

Line 13: “in older adults living at their homes (N=245, M=74, SD=6,654)” From this fragment it is not clear what M and SD mean. I guess it is mean age and standard deviation of age, but it should be clarified.”

Lines 15-16: “a human in charge of the users' (health) data significantly altered the attitude toward using an assistive technological system” After reading the paper I cannot understand what you mean by “a human in charge of the users' (health) data” I didn’t find such question/variable in the paper, so I think that these should not be emphasized in the abstract.

Line 41-42: “(see also AUTHOR 2020)” It needs clarification and completion in the reference section. Perhaps it was made to hide the author’s name for double blind review.

Lines 119-126: "Imagine you are learning about an assistive technology system that seems appropriate for you and your household. a) Would you be willing to spend money on it? b) Would you be willing to change your daily habits (e.g., mealtimes) for it? c) Would you be willing to change your accustomed environment (e.g., floor coverings) for it?" Those participants who showed a willingness to change on all three questions were labelled "technology optimists". Those who indicated an unwillingness to change on at least one of the questions  (to a maximum of all three questions) were labelled "technology pessimists".

 How the answers to these questions were formulated? Binary yes-no? It should be explained.

Lines 132-139: Description of  Figure 1. “Methodological conceptualization of a potential interrelation between technology acceptance of the questioned persons (they are split into "technology optimists" and "technology pessimists" as indicated by the grey columns) and socio-economic variables (as captured by the covariates: sex, age, education, income), social variables (captured by the variables: cohabitation, social integration), QoL variables (captured by the variables: Quality of Life, technology impact on QoL), health variables (captured by the variables: self-rated health, physical limitations, falls), attitude toward invasive technology, and attitude toward ageing (created in “keynote” V8.1) Socio-economic variables include questions about sex, age, education, and income.”

There is an unnecessary repetition concerning socio-economic variables, I marked it in bold.

Section 2.4. Please indicate clearly what multivariate method you use. From the text provided one can conclude that you use regression analysis, but what type – logistic as your dependent variable is binary?

Line 180 “2.4. Multivariate statistics”. I suggest changing the title of these paragraphs for more appropriate referring to regresion.

Lines 187-189: “education as continuous variable encoded as: 1: Compulsory school, 2: Vocational or commercial school, high school, university-related education, university degree”.  What do you mean by “education as continuous variable?” The division into two categories is a bit surprising here: category 2 includes both vocational school and university degree, persons with very different levels of education.

Lines 231-232: “About one-third (37,0%, N = 230) of the people could imagine their medical parameters such as blood pressure or sugar measured by a chip. The collected data is regularly transmitted to their doctor” Is the data transmitted or would be transmitted if the persons agree to use the chip?

Paragraphs 3.1-3.5. In your description there are different sample size values (N). I guess this is because some respondents didn't answer certain questions. If this is the reason, I think it is worth including this information for the reader.

Line 273: “2.4. Multivariate statistics” and 3.6. Multivariate Statistics” This part shows the results of regression analysis (I guess logistic one). So I suggest changing the title of these paragraphs for more appropriate.

Line 297, line 300: “explaining variables” better: exploratory variables

Lines: 275-276 “Figure 2. presents an overview of the results of multiple regressions between "technology optimists and "technology pessimists"”. I would not say that this picture presents the results of regressions, in my opinion the figure is obscure and the only thing indicated by it is if a parameters of regressions are significant. Are these regressions really “between optimists and pessimists”? I cannot understand it. My guess is that these are regressions evaluated separately for optimists and pessimists where the dependent variable is the attitude toward the use of a technological system and other considered variables. If so you should clarify it in the paper.

Tables 1-4. You write that you consider two groups "technology optimists and "technology pessimists” and you confirm it in Figures 1 and 2, as I suppose. But there are not results in the tables for optimists and pessimists separately. My question is if these regression results are for the whole sample? And if so how it is connected to your research idea for dividing into optimists and pessimists? Anyway, you must explain it thoroughly in the article how the regression was constructed and calculated.

Lines 282-283: “We found that a specific event, that made a person feel older also changed their attitude toward using a technological system positively (see table 1)”

The regression coefficient by “Feeling older since a certain event” is negative, odds ratio is below 1 but you interpret it as a positive impact.

The same holds for coefficients by sensor and microchip (tables 2-3).

The interpretation needs revision. Maybe the misunderstanding is due to your independent variables coding (not shown in the text, so I cannot judge).

To conclude: the presentation of the results needs many clarifications. The results must be corrected and explained better otherwise the article cannot be published

Author Response

(The authors gave the same response as above.)

Round 2

Reviewer 1 Report

Comments and Suggestions for Authors

The authors appropriately reacted on the criticism and motivated their viewpoint. As I am still not fully convinced by the research design / approach I left some of the criteria "average", but this is a discussion on the level of different scientific disciplines/perspective. The changes and adaptions are - as mentioned - appropriate, therefore I do not have objections to publication. 

Author Response

Thank you. We added another paragraph and hope to stimulate further academic debate.

Best regards,

Ulrike

Reviewer 3 Report

Comments and Suggestions for Authors

In my review I asked about the negative regression coefficients by variables indicating feeling older, sensor and microchip. The Authors' answer that this negative values are caused by the way the variables were encoded explained the interpretation to me but it is not incluced in the text. To clarify the intepretation to the reader I suggest to put the explanation (the answer in the paragraph starting with "The significant negative association between...") in the paper when interpreting the results of the regression. 

Author Response

Thank you. We added the respective paragraph as you requested and think that this does indeed increase the clarity of the paper. It is marked in yellow and was inserted as of line 285.

Best regards
